# Phase Separation and Mechanical Forces in Regulating Asymmetric Cell Division of Neural Stem Cells

**DOI:** 10.3390/ijms221910267

**Published:** 2021-09-24

**Authors:** Yiqing Zhang, Heyang Wei, Wenyu Wen

**Affiliations:** Department of Neurosurgery, Huashan Hospital, The Shanghai Key Laboratory of Medical Epigenetics, Institutes of Biomedical Sciences, State Key Laboratory of Medical Neurobiology and MOE Frontiers Center for Brain Science, School of Basic Medical Sciences, Fudan University, Shanghai 200032, China; yiqingz36@163.com (Y.Z.); 20111510041@fudan.edu.cn (H.W.)

**Keywords:** asymmetric cell division, mechanical force, myosin flow, neural stem cell, phase separation, polarity cue

## Abstract

Asymmetric cell division (ACD) of neural stem cells and progenitors not only renews the stem cell population but also ensures the normal development of the nervous system, producing various types of neurons with different shapes and functions in the brain. One major mechanism to achieve ACD is the asymmetric localization and uneven segregation of intracellular proteins and organelles into sibling cells. Recent studies have demonstrated that liquid-liquid phase separation (LLPS) provides a potential mechanism for the formation of membrane-less biomolecular condensates that are asymmetrically distributed on limited membrane regions. Moreover, mechanical forces have emerged as pivotal regulators of asymmetric neural stem cell division by generating sibling cell size asymmetry. In this review, we will summarize recent discoveries of ACD mechanisms driven by LLPS and mechanical forces.

## 1. Introduction

Asymmetric cell division (ACD), which generates daughter cells with distinct fates, is a fundamental mechanism for cell diversity and the normal development of multicellular organisms. The *Drosophila* neural stem cell neuroblast (NB) is regarded as an important model system for understanding the ACD of stem cells and progenitors. Through ACD, *Drosophila* NBs each produce a larger daughter cell that inherits the characteristics of NBs and can continue asymmetric division and a smaller ganglion mother cell (GMC) which undergoes terminal differentiation to generate two neurons or glial cells (Figure 1) [1]. Importantly, the molecular mechanisms and signaling pathways that regulate the self-renewal and differentiation of NBs are evolutionarily conserved from *Drosophila* to mammals [2].

At the metaphase of dividing *Drosophila* NBs, the PAR complex and its related proteins are distributed on the apical cortex of the cell, whereas cell fate determinants and their adaptor proteins are localized at the basal cortex [2,3,4]. Instead of being uniformly distributed on the apical or basal half of the cell cortex, these proteins form crescent structures at the opposite poles of the cell, respectively (Figure 1) [5,6,7,8,9]. Meanwhile, actomyosin-mediated contraction of actin filaments beneath the cell cortex and dynein-induced cortical pulling on astral microtubules (MTs) load mechanical forces to the cell, thereby changing its shape and behavior, spindle orientation and positioning, and, finally, generating sibling cells with distinct size and function [10,11]. Recently, the liquid–liquid phase separation (LLPS) of biomolecules has been found to play an important role in the organization and regulation of various cellular membrane-less organelles [12,13,14,15]. Several studies reported that the apical and basal crescents in dividing NBs are formed by phase separation of certain proteins [16,17,18].

In this review, we will briefly summarize the basic mechanism of intrinsic ACD using *Drosophila* NBs as the paradigm, and then we highlight recent findings of biomolecular LLPS and mechanical forces in regulating ACD of *Drosophila* NBs.

## 2. ACD of *Drosophila* Neuroblasts

Drosophila neurogenesis occurs in two stages, the embryo stage and the second instar larval stage, each of them relying on rounds of asymmetric and symmetric division of NBs to generate various types of neurons [1]. Extrinsic and intrinsic mechanisms function coordinately to regulate the ACD process of NBs. Even though stem cells or progenitors divide symmetrically, different microenvironments around the two identical daughters induce them to adopt distinct fates. Whereas intrinsically, cell fate determinants are asymmetrically distributed at the cortex to set up cell polarity at metaphase. Subsequently, spindle dependent and independent mechanisms act cooperatively to ensure the unequal segregation of cell fate determinants and the generation of two distinct sibling cells [19].

### 2.1. Asymmetric Protein Localization during ACD

The conserved PAR complex, containing Bazooka (Baz, Par3 in mammals), Par-6, and atypical protein kinase C (aPKC), resides at the top hierarchy of cell polarity regulators and is essential for the asymmetric localization of cell fate determinants and the establishment of polarity. When an NB undergoes asymmetric division, PAR proteins, which uniformly diffuse on the membrane or cytoplasm during interphase, start to assemble and concentrate at the apical cortex, where they are most concentrated (crescents) at metaphase. The apical PAR complex induces the apical anchoring of the adaptor protein Inscuteable (Insc), which further recruits the hub protein partner of Inscuteable (Pins) [8,20,21]. Interestingly, cell fate determinants including the Notch inhibitor Numb, the transcription factor Prospero (Pros), *pros* mRNA, the translational repressor Brain Tumor (Brat), and the double-stranded RNA binding protein Staufen (Stau), as well as their adaptors Miranda (Mira) and Partner of Numb (Pon), are apically localized together with the PAR complex initially [1]. Of note, aPKC in the PAR complex is locked in an inactive state. AuroraA-mediated activation of aPKC leads to phosphorylation of Numb, Pon, and Miranda at their cortical binding domains, which results in the release of these proteins from the apical cortex together with their binding partners [22,23,24]. At the same time, activated aPKC can phosphorylate Baz/Par3 and lead to its dissociation from the Par-6–aPKC subcomplex. Thus the highly enriched PAR crescent begins to expand on the apical cortex from anaphase and finally exhibits uniform cellular distribution after cytokinesis (Figure 1).

On the other hand, apically excluded Numb is then targeted to the basal cortex by binding to Pon, forming a concentrated crescent at metaphase (Figure 1). Similarly, apically excluded Mira is transferred to the basal cortex, and the stable basal anchoring of Mira is facilitated by its mRNA [25]. Pros, Brat and Stau (together with *pros* mRNA) are also targeted basally through Mira via its cargo binding domain [26]. Guided by the apical PAR complex, the basal crescent starts to disassemble from anaphase.

### 2.2. Generation of Distinct Sibling Cells

The transient apical-basal cell polarity is intimately bound with cell fate control. After cytokinesis, the apical daughter that inherits the PAR complex adopts an NB fate. Cell fate determinants are inherited by the basal GMC, in which Numb promotes its differentiation mainly through antagonizing Notch signaling, Mira is degraded in the cytoplasm, whereas Pros enters the nucleus of the GMC, where it preferentially inhibits the expression of cell cycle regulatory genes and activates the expression of cell differentiation genes to ensure neural terminal differentiation [7]. Through binding to the 3’ UTR of *pros* mRNA, Stau enters the nucleus of GMC after asymmetric division [27].

To ensure the uneven segregation of these apical and basal proteins in sibling cells, the cleavage plane needs to be perpendicular to the cell polarity axis, a process that relies on several evolutionarily conserved force-generating protein complexes. The apically localized hub protein Pins can form a complex with Gαi and Mushroom body defect (Mud) [21]. Cortically localized Gαi enhances the cortex anchoring of Pins and also cooperates with Mud to transform Pins from the auto-inhibited state to the open conformation. On the other hand, Mud is bound to the minus-end-directed Dynein–Dynactin motor complex on the astral microtubules of the spindle. Thus, Pins forms a super-complex with the PAR–Insc complex at the apical cortex and the Gαi–Mud–Dynein–Dynactin complex on the astral microtubules, which then pulls the mitotic spindle along with the cell polarity [4]. Moreover, polarity cues are necessary for cleavage furrow localization by regulating the location of actomyosin, a key component of the contraction ring [28,29]. In the asymmetric division of stem cells, mislocalization of the cleavage furrow leads to aberrant segregation of cell fate determinants, resulting in changes in cell fate and cell behavior, which have been closely correlated with developmental disorders and other diseases [3,30]. In addition, spatiotemporal regulation of actin-cytoskeleton is also a key mechanism for generating sibling cell size asymmetry via biased cortical expansion [11].

## 3. LLPS and Asymmetric Protein Localization during ACD of NBs

As a physical process, LLPS occurs when a supersaturated solution spontaneously separates into two stably coexisting phases, a condensed phase with selectively recruited components demixing from the original sparse phase, just like oil droplets demixing from water, and the two phases are constantly exchanging the common components [31]. Since 2009, accumulating evidence has shown that biomolecules (e.g., proteins and nucleic acids) can undergo LLPS in the cell, which is now regarded as an important mechanism for the autonomous assembly of membrane-less organelles, including various RNA granules (P granules, stress granules), centrosomes, nucleolus, and signaling assemblies [13,15,32,33,34,35,36].

### 3.1. LLPS in Cells

By selectively recruiting certain biomolecules and excluding the others, LLPS enables the selected biomolecules to execute unique biological functions. As the concentration of biomolecules in the liquid phase is significantly higher than that in the surrounding environment, LLPS provides an efficient and economical way to increase their local concentration within a micrometer-scale compartment to speed up the reaction. On the other hand, without the physical separation by membranes, biomolecules in the condensed phase are highly dynamic and able to freely exchange between the liquid phase and its surroundings, and thus are sensitive to environmental changes, cell signals, or posttranslational modifications which promote or inhibit the LLPS process. Abnormal phase separation of biomolecules may cause their phase transition into intracellular aggregates without dynamics, which are causally associated with a variety of human diseases, including neurodegenerative diseases, cancer, cataract, and aging-related diseases [12,13,37,38,39].

Detailed biochemical analysis reveals that the main driving force for LLPS is multivalence, which can be achieved in two ways. One is mediated by the non-specific weak association of the flexible unstructured intrinsically disordered regions (IDRs) or low-complexity domains (LCDs) [40,41,42]. The other way involves specific recognition between nucleic acids and/or proteins that contain multiple folded domains [14,43,44,45]. The LLPS property of certain biomolecule(s) largely depends on the concentration and property of biomolecule(s), as well as the surrounding solutions [46]. The external physical environment can dynamically regulate LLPS by changing the affinity between the polyvalent molecules, including temperature, pH, salt concentration and pressure [14].

### 3.2. LLPS-Mediated Basal Localization of Numb and Pon in Dividing NBs

Though the concentrated apical and basal crescents during ACD of Drosophila NBs have been observed for more than 20 years, it was only recently reported that such crescents are indeed clustered protein condensates autonomously formed via LLPS, such as the Numb–Pon and PAR complexes [16,17]. Shan et al. provided in vitro and in vivo evidence showing that the cell fate determinant Numb undergo LLPS upon binding to its adaptor Pon, which might be the potential driving force for the basal targeting of Numb–Pon (Figure 2A) [16]. Additionally, this study firstly brought the concept of LLPS into ACD. The phosphorylated tyrosine binding (PTB) domain of Numb was found to specifically recognize the N-terminal repeating motifs of Pon in a non-canonical mode (Figure 2A), which led to the formation of a heterogeneous Numb–Pon interaction network that presented as condensed liquid droplets in vitro. Though the recombinant Numb PTB or Pon alone existed as a homogeneous sparse phase in the same buffer solution, mixing the two proteins led to autonomous condensation and demixing of the Numb–Pon complex from the sparse phase. These spherical protein droplets rapidly coalesced to form larger ones, a characteristic of LLPS. Similar puncta-like structures were observed in the cytoplasm when Numb and Pon were co-expressed in Hela cells. Fluorescence recovery after photo-bleaching (FRAP) analysis of the fluorescently labeled proteins showed that both Numb and Pon in the droplets or puncta possessed high dynamic property and were constantly shuttling between the condensed phase and surroundings rapidly, with a speed comparable to that of the corresponding proteins in the crescent of dividing Drosophila NBs [47]. Transgenic flies expressing LLPS deficient Pon mutant displayed diffusion of endogenous Numb from the basal cortex, which further caused an abnormal ACD process and a tumor-like phenotype with over-proliferating NBs [16]. Interestingly, LLPS of the Numb–Pon complex was a reversible process. Pre-formed Numb–Pon liquid droplets could be reversed to the sparse phase by a CDK1-mediated phospho-Pon peptide which competitively binds to Numb PTB [16]. It will be interesting to investigate the potential regulatory effect of CDK1 activity on the LLPS property and localization of the Numb–Pon complex.

### 3.3. LLPS-Mediated Apical Localization of the PAR Complexes in Dividing NBs

In addition to basally concentrated cell fate determinants, it has been demonstrated that the apical crescent is also a consequence of LLPS-mediated local condensation of PAR proteins (Figure 2B) [17]. From the prophase, Baz/Par3, Par-6 and aPKC were observed as scattered puncta on the apical membrane of NBs, which grew into larger ones and appeared as a crescent at metaphase from the apical-basal view. The addition of 1,6-hexanediol (1,6-HD), an aliphatic molecule widely used to disrupt liquid condensates driven by hydrophobic interactions [48], led to the diffused localization of the apical PAR complex as well as the basal adaptor Mira in larval brains in a dose-dependent manner. Crescents of PAR proteins and Mira reappeared after removing 1,6-HD [17], revealing the dynamic and reversible nature of these protein condensates. The formation of such dynamic PAR condensates is mainly driven by oligomerization of Baz/Par3 via its N-terminal domain (NTD) (Figure 2B). Supporting this notion, the overexpression of Baz/Par3 in a non-polarized Drosophila S2 cell induced the formation of cortical Baz/Par3 patches, which had a liquid-like property [49]. Par-6 could be enriched in the Baz/Par3 condensates by binding to Baz/Par3 PDZ3 via its PDZ binding motif (PBM), and Par-6 could self-associate through its PB1 domain, which further enhances the multivalence of the Baz/Par3–Par-6 complex as well as its LLPS property (Figure 2B) [17]. As LLPS is a concentration-dependent process, to avoid the artificially increased LLPS property through overexpressing exogenous proteins in transgenic flies, Shan et al. constructed knock-in flies expressing endogenous levels of Baz/Par3 wild-type protein and various mutants to examine the effect of LLPS in polarized protein localization as well as the ACD process. LLPS-deficient Baz/Par3ΔNTD mutants exhibited significant cytoplasmic diffusion of both apical Baz/Par3–Par-6–aPKC proteins and basal protein Mira, which led to impaired ACD process and a smaller brain phenotype. However, when Baz/Par3ΔNTD was fused with FUS LCD, a fragment known to have a strong LLPS property [39], the chimeric mutant largely rescued the normal distribution of apical and basal proteins, as well as the brain size, demonstrating the essential role of Baz/Par3–Par-6 LLPS in regulating cell polarization and ACD of Drosophila NBs.

aPKC, the only kinase in the PAR complex, can also be recruited into the Baz/Par3–Par-6 condensate by its PB1 domain interacting with Par-6 PB1 domain, its kinase domain interacting with the conserved region3 (CR3) of Baz/Par3, and its C-terminal PBM recognizing the PDZ2 of Baz/Par3 (Figure 2B) [50,51,52]. Interestingly, aPKC in the Baz/Par3–Par-6 condensates is inactive, though aPKC can phosphorylate Par3 CR3 [53], and the phospho-mimetic Baz/Par3 mutant exhibited a significantly weakened LLPS property [17]. It is plausible that aPKC is recruited and transported to the apical membrane via Baz/Par3–Par-6 condensates, with the suppressed kinase activity. When aPKC reaches the apical cortex, cell-cycle regulator(s) induce(s) its activation, which subsequently leads to phosphorylation of Baz/Par3, as well as of cell fate determinants and their adaptors. One potential regulator might be Cdc42, as aPKC was reported to cycle between a Baz/Par3-bound pool with low activity and a Cdc42-bound pool with high activity during the polarization of C. elegans embryo [54,55]. The phosphorylation of Baz/Par3 results in the disassembly of the PAR condensates, which further releases active aPKC. Phosphorylated cell fate determinants and their adaptors are excluded apically and then concentrated at the basal cortex to set up the apical-basal polarity. It is assumed that a balance between apical condensation of the Baz/Par3–Par-6 complex (together with inactive aPKC) and activated aPKC-mediated disassembly of the Baz/Par3–Par-6 condensate. The outcome is that the apical condensation of the PAR complex reaches the peak at metaphase and starts to disassemble from anaphase. Such polarization and depolarization processes may also be regulated by the actin cytoskeleton [56]. Apical-directed cortical flow accelerates the apical condensation of aPKC at metaphase, whereas at anaphase onset, the cortical flow changes its direction towards the cleavage furrow and promotes the disassembly of apical aPKC patches [56].

### 3.4. LLPS-Mediated Mitotic Implantation of Pros in Dividing GMCs

After cytokinesis, the transcription factor Pros enters the GMC nucleus to promote its differentiation [57,58]. Recently, Liu et al. showed that Pros drives irreversible terminal neuronal differentiation by regulating heterochromatin domain condensation and expansion in an LLPS-dependent manner (Figure 2C) [59]. Pros was found to undergo LLPS in vitro and in vivo through self-association through its N7 motif (Figure 2C). LLPS of Pros enabled its retention at histone H3 Lys9 tri-methylation (H3K9me3) heterochromatin regions of chromosomes in mitotic GMCs, where it recruited and concentrated the H3K9me3 “reader” heterochromatin protein 1 (HP1) into the condensed phase via its N-terminal domain (Figure 2D), thus driving the condensation and expansion of the H3K9me3^+^ heterochromatin regions in the newly generated neurons. After HP1 condensation, Pros, together with a portion of HP1, detached from the H3K9me3^+^ heterochromatin regions and translocated to its target gene loci, where Pros and HP1 acted cooperatively to silence Pros target genes permanently to drive cell-cycle exit and terminal neuronal differentiation [59]. Pros mutants that exhibited impaired LLPS ability prevented Pros from being retained on chromosomes and thus resulted in compromised terminal differentiation. The above phenotype could be effectively rescued by replacing the N7 motif with another IDR protein capable of LLPS. Interestingly, though the recombinant N7-containing Pros fragment and HP1a co-phase separate in vitro, the Pros condensates and HP1a condensates do not coalesce in vivo [59], suggesting the existence of unknown regulating mechanism(s). Moreover, it is plausible that the basal distribution of Pros in dividing NBs might also be driven by its phase separation, together with the Mira dimer via its coiled-coil domain (Figure 2A).

## 4. Mechanical Forces Regulating ACD

Sibling cells generated by ACD of Drosophila NBs have markedly different sizes and components (e.g., polarity proteins and cell fate determinants), thus adopting distinct fates. Such asymmetry can be achieved by cooperative mechanisms in spindle-dependent and independent ways. A spindle-dependent mechanism highly depends on the orientation and positioning of the mitotic spindle, whereas a spindle-independent mechanism involves unequal cortical expansion and correct location of the cleavage furrow of NBs, which is determined by the distribution of non-muscle Myosin II (referred to as Myosin) [11].

### 4.1. Polarity Cue-Regulated Spindle Orientation

During ACD of Drosophila NBs, in addition to the central Pins–Gαi complex that provides a cortical cue for positioning and orientation of the mitotic spindle via the Mud–dynein–dynactin machinery, other regulators have recently been identified. The junctional scaffold Canoe (Afadin in mammals) was found to be a component of the apical Insc–Pins (LGN)–Gαi–Mud (NuMA) super-complex, in which it regulates the spindle orientation by recruiting Mud to the cortex and thus activating the Pins-Mud-dynein pathway in a RanGTP-dependent manner (Figure 3A) [60,61,62]. Similarly, Afadin regulates the apical-basal spindle orientation during cell division in developing renal tubules [63]. Two independent investigations revealed that the Hippo pathway kinase Warts is involved in this process by phosphorylating both Canoe and Mud to promote Pins-Mud complex-mediated spindle orientation [64,65]. Intriguingly, a recent structural analysis suggested that Afadin binds to LGN in a manner that resembles the Insc–LGN and NuMA–LGN interactions [21,66]. The three components within this complex, Insc, NuMA and Afadin, all interacted with the TPR domain of LGN at the same target binding surface (Figure 3A), which seems contradictory to their physiological function that act cooperatively to mediate spindle orientation.

Recently, the cytosolic tail of the adhesion molecule E-cadherin has been found to act as a cortical cue for spindle orientation by recruiting LGN to cell-cell contacts in MDCK cells [67]. Guided by this spatial information, NuMA was targeted to cell–cell adhesions together with astral microtubules by locally competing for LGN from E-cadherin during mitosis and thus oriented the mitotic spindle [67,68]. As is the case for Afadin, E-cadherin is bound to the same target binding pocket in the TPR of LGN. It remains elusive why so many proteins competitively interact with LGN TPR but exert a cooperative role in mitotic spindle orientation.

### 4.2. Myosin Flows Regulated by Polarity and Spindle Cues

Sibling cell size asymmetry mainly results from the biased cortical expansion and controlled cleavage furrow positioning, both of which rely on the dynamic localization of actomyosin and modulation of its contractility [11]. In dividing Drosophila NBs, the intrinsic polarity cue Pins–Gαi has been found to guide the correct localization of myosin spatiotemporally, thus controlling the cleavage furrow position and daughter cell size independent of the mitotic spindle [28,29]. Tsankova et al. recently found that the Rho kinase (Rok) and protein kinase N (Pkn) function sequentially to regulate biased myosin activity and localization in response to Pins in dividing Drosophila NBs (Figure 3B) [69]. Myosin is a substrate of Rok, and Rok-mediated phosphorylation of myosin induces its activation. At the early metaphase, Pins recruits Rok apically, which further concentrates the activated myosin at the apical cortex. Following the apical enrichment of Pkn (via Pins) at the late metaphase, Rok activity is downregulated, and active myosin is dephosphorylated and timely excluded from the apical cortex [69]. As a consequence, the translocation of myosin, which is originated from polarity cues, results in a cortical myosin flow heading the basal cortex and consequent cortical expansion of the apical cortex, which contains fewer myosin filaments with weaker contractile forces [70].

Shortly after the start of the above spindle-independent, basally directed myosin flow (about one minute), another apically directed myosin flow is generated on the basal cortex, which is triggered by the central spindle pathway (Figure 3B) [71]. Microtubules from the central spindle contact the equatorial cortex, leading to localized activation of the small GTPase Rho1 via delivery of the centralspindlin complex at the lateral cortex, which subsequently results in local enrichment and activation of myosin where the cleavage furrow is positioned [71]. Such local enhancement of myosin activity then triggers the basal cortical flow, as the intrinsic contractile property of myosin drives it to move towards the highest myosin density [72]. Consequently, the spindle cue clears myosin from the basal cortex and thus results in the accumulation of myosin at the cleavage furrow. Both the apical and basal cortex expansions are induced by clearance of myosin and relieved actomyosin contractile tension on the apical and basal cortex, respectively. However, the relatively prolonged expansion of the apical cortex results in a larger apical daughter and a smaller basal daughter. A follow-up study suggested that besides the myosin-mediated constriction, intracellular hydrostatic pressure further enhances cortical expansion at the apical cortex at anaphase onset [73].

Taken together, through spatiotemporal polarity and spindle cues, Drosophila NBs establish successive apical and basal cortical myosin flows, relocate myosin to the lateral cortex at anaphase onset, and, thus, determine the cleavage furrow site and enable biased cortical expansion, finally building up physical asymmetry in dividing NBs.

## 5. Conclusions and Perspectives

Neural stem cells generate sibling cells with distinct sizes, components, and fates through asymmetric division, which can be achieved through interplayed intrinsic mechanisms involving asymmetric localization of polarity cues, regulated spindle orientation, generation of myosin-mediated actin flows, and cleavage furrow positioning. Recent studies have suggested that LLPS is a driving force of the autonomous assembly of the highly enriched protein crescents beneath the apical and basal cortex in dividing Drosophila NBs. Through domain recognition or oligomerization-mediated multivalent interactions, the Baz/Par3–Par-6–aPKC complex and the Numb–Pon complex (likely the Miranda–Pros–Staufen–Brat complex as well) form apical and basal protein condensates spatiotemporally, thus establishing the cell polarity and providing polarity cues for the following ACD-related processes [74]. Compared with the classical membrane anchoring or clustering mechanism, the LLPS theory presents prominent advantages, the most important one being the high dynamics property of protein condensates, which is assumed to be essential for the fast response to cell-cycle signals to assemble or disassemble the cell polarity cues. Moreover, as RNAs with large disordered regions also have high potency for LLPS [75,76], it is plausible that pros mRNA and mira mRNA may further enhance the LLPS property of the Mira-related basal proteins, facilitating their asymmetric localization and segregation during ACD of NBs [25,77].

To generate two distinct sibling cells, the ACD process is accompanied by various forms of mechanical forces arising from precise rearrangements of the actin/microtubule cytoskeleton. Some mechanical forces are highly correlated with polarity cues at the local membrane regions. In this review, we summarize the assembly of the force generator (Pins/Canoe/Mud) for spindle orientation and spatiotemporal generation of myosin flows for biased cortical expansion and cleavage furrow positioning, both of which are regulated by the polarity cue Pins. As Pins is a component of the apical PAR-related super-complex, whether LLPS of polarity cues or other biomolecules plays a regulatory role in the generation and function of intrinsic mechanical forces is an intriguing question. More broadly, whether LLPS and mechanical forces interplay with each other is an interesting research direction.

Several studies have suggested the involvement of biomolecular LLPS in driving the organization of the cytoskeleton. In the presence of Filamin, short actin filaments were recently found to undergo phase separation, whereas long filaments were prone to form gel-like structures via phase transition [78]. Spindle matrix component, BuGZ, phase separated via its C-terminal IDR to promote MT polymerization and assembly of spindle matrix [79]. The MT-binding protein Tau underwent LLPS via its IDR under physiological conditions, whereas neurodegenerative disease-associated hyper-phosphorylation promoted Tau phase transition into gel-like aggregates incapable of MT binding, finally causing neuronal cell death [80,81,82]. On the other hand, the formation of the condensed crescents during ACD is intimately connected with the mechanical force caused by cortical flows. In dividing Drosophila NBs, actin cytoskeleton-dependent cortical flows facilitated the assembly of the apical aPKC cap at metaphase and its disassembly at anaphase onset [56]. During embryonic polarization in C. elegans, actomyosin contractility and the resulting cortical forces stimulate PAR proteins clustering on the cortex [54].

One can assume that biomolecular LLPS and mechanical forces cooperatively regulate ACD of neural stem cells and cell division in general. Many related questions need to be answered. Is LLPS a potential mechanism to explain why so many proteins competitively interact with Pins but can still function collaboratively? Does the attachment of the highly enriched protein condensates to the inner surface of the cell cortex change membrane curvature to generate an asymmetry? Does this process lead to a rearrangement of the actin cytoskeleton beneath the membrane and consequently changes in mechanical forces? Is the assembly of other actin/MT cytoskeleton-related apparatuses driven by LLPS? Further investigation in this direction will significantly broaden our understanding of the regulating mechanisms of neural development and diseases.

## Figures and Tables

**Figure 1 ijms-22-10267-f001:**
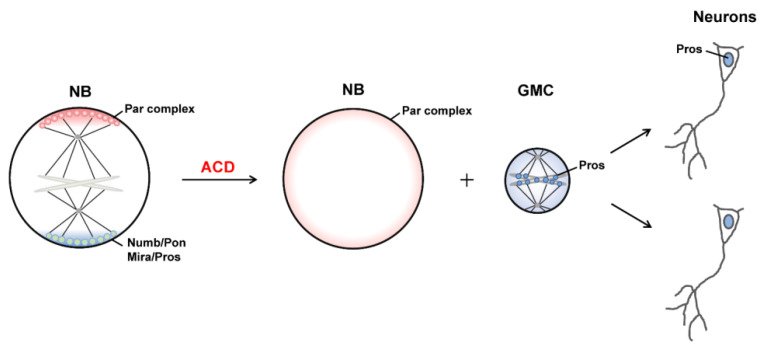
Schematic diagram showing the ACD process of Drosophila NBs. An NB divides asymmetrically to generate a larger NB and a smaller GMC through asymmetric inheritance of cell fate determinants. GMC undergoes another round of symmetric division to produce two neurons. Polarity proteins and cell fate determinants exhibit cell-cycle-dependent localization.

**Figure 2 ijms-22-10267-f002:**
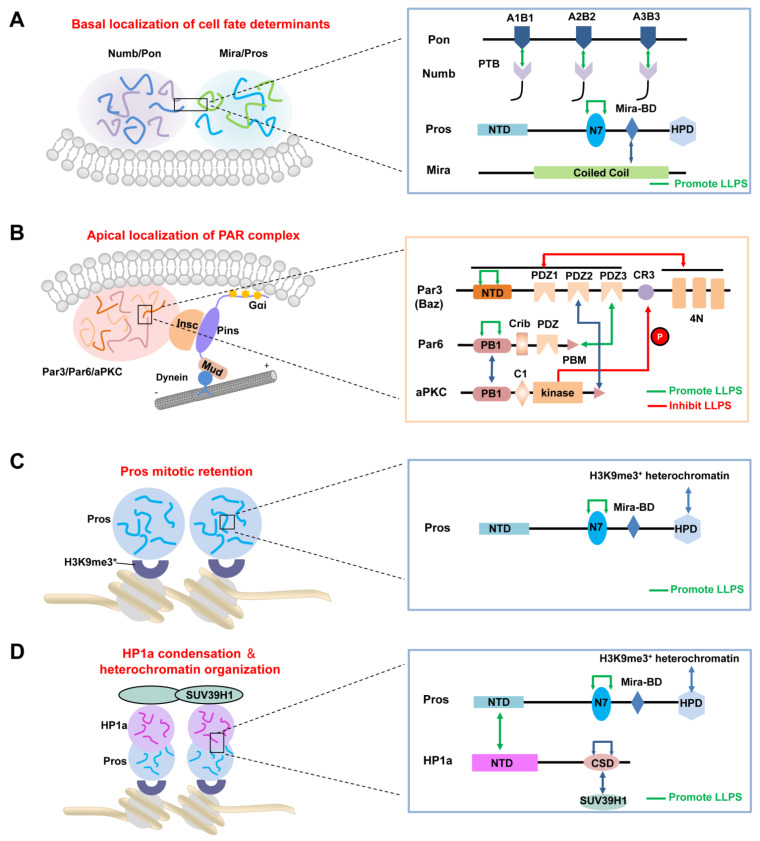
Polarity cues, including the PAR complex, Numb–Pon complex, and Pros with its various binding partners, play essential roles in different stages of asymmetric division of *Drosophila* NBs via LLPS. (**A**,**B**) The basal (**A**) and apical (**B**) protein condensates during ACD of NBs. (**C**) The mitotic retention of Pros at H3K9me3^+^ heterochromatin regions of chromosomes in mitotic GMCs via its LLPS. (**D**) Pros mediates HP1 condensation and thus drives the condensation and expansion of the H3K9me3^+^ heterochromatin regions in the newly generated neurons. The above proteins undergo LLPS spatiotemporally through multivalent interactions, which involve specific domain-domain recognition and self-oligomerization.

**Figure 3 ijms-22-10267-f003:**
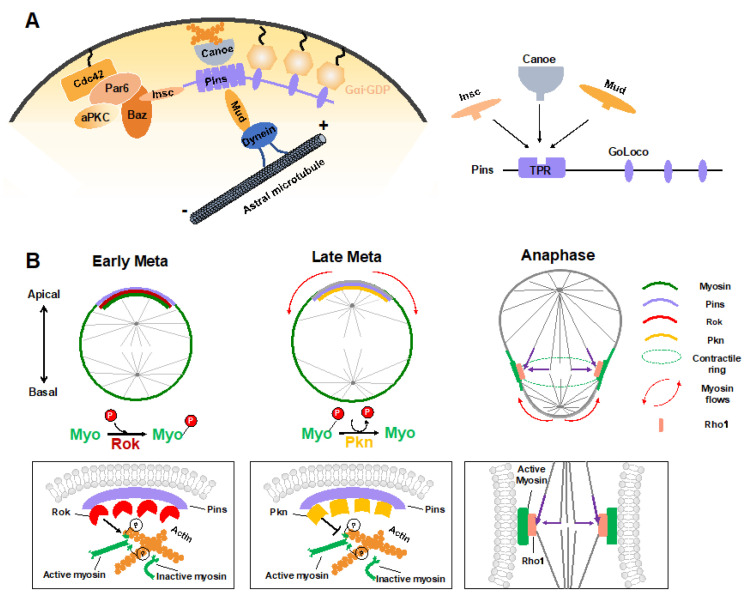
Mechanical forces in regulating ACD of *Drosophila* NBs. (**A**) Apical polarity cues Pins and Canoe mediate the assembly of force generators of spindle orientation via guiding the cortical attachment of the Mud–Dynein complex. Intriguingly, Insc, Canoe, and Mud competitively bind to Pins, even though they function cooperatively in the polarity-guided spindle orientation process. (**B**) Polarity cues and the mitotic spindle regulate spatiotemporal myosin flows to determine biased cortical expansion and cleavage furrow positioning to generate sibling cell size asymmetry. Rok activates myosin through phosphorylation and mediates its cortical localization before mitosis. Pins recruits Rok apically at early metaphase and thereby enriches active myosin at the apical cortex. Subsequently, Pins recruits Pkn apically at late metaphase, leading to timely apical myosin clearance by inhibiting myosin activity. The relief of myosin contraction at the apical cortex leads to its cortical expansion. At anaphase onset, the spindle-mediated accumulation of active myosin at the lateral membrane (via Rho1) promotes the basal myosin clearance and basal cortical expansion. The lateral membrane site with enriched myosin determines the cleavage furrow position.

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
