# Peer review of "Phase Separation and Mechanical Forces in Regulating Asymmetric Cell Division of Neural Stem Cells"

_ijms, 2021, doi:10.3390/ijms221910267_

Round 1

Reviewer 1 Report

The authors in this review article elaborated on the role of phase separation and mechanical forces in the model system of the Drosophila neural stem cell neuroblast (NB) for the understanding of the Asymmetric cell division (ACD).

They discussed functional mechanisms with their possible physiological relevance and pathologies. This review would be of interest to readers in the biology of cell division regulation with the new perspective of the novel LLPS mediated regulation. The authors have covered all the relevant progress and studies in this review.

Authors have given a very clear explanation and history of the LLPS and mechanism discovered by Shan et al. under the title paragraph “LLPS-mediated basal localization of Numb and Pon in dividing NBs” which is very insightful. It is interesting to find out by methodically increase complexity to determine how known modulators of LLPS influence droplet formation and their characterization. A comment on the role of the RNA in this LLPS process could be useful and whether RNA affects here in this particular case!

Reviewer 2 Report

Drosophila neuroblasts (NBs) are a classic model system for studying asymmetric cellular division (ACD). This review presents an overview of ACD in Drosophila NBs, and discusses the recently defined contributions of liquid-liquid phase separation (LLPS) and mechanical forces to ACD. Finally, the authors opine on future mechanistic studies on the role of LLPS and mechanical forces in ACD. This review is informative and the text well organized, it will be of great value for researchers interested in neural stem cells. However, as it stands, many grammar/syntax errors detract from the content. Rather than Iisting all those, I attached a file with suggestions. I also have specific comments.

Figure 1 looks a little too dense. I would suggest splitting it up into two figures. One focusing on basic mechanisms related to ACD, and one focusing on LLPS during ACD. In the first one it would be helpful to show more clearly the cell size difference after division.

Line 114: Myosin is not really part of the cytoskeleton per se. Are the authors specifically referring to actin? If yes, I would suggest changing the term accordingly.
